# High-Throughput Screening Assay for Detecting Drug-Induced Changes in Synchronized Neuronal Oscillations and Potential Seizure Risk Based on Ca^2+^ Fluorescence Measurements in Human Induced Pluripotent Stem Cell (hiPSC)-Derived Neuronal 2D and 3D Cultures

**DOI:** 10.3390/cells12060958

**Published:** 2023-03-21

**Authors:** Hua-Rong Lu, Manabu Seo, Mohamed Kreir, Tetsuya Tanaka, Rie Yamoto, Cristina Altrocchi, Karel van Ammel, Fetene Tekle, Ly Pham, Xiang Yao, Ard Teisman, David J. Gallacher

**Affiliations:** 1Global Safety Pharmacology, Preclinical Sciences and Translational Safety, Janssen R&D, A Division of Janssen Pharmaceutica NV, B-2340 Beerse, Belgium; 2Elixirgen Scientific, Incorporated, Baltimore, MD 21205, USA; 3Healthcare Business Group, Drug Discovery Business Department, Ricoh Company Ltd., Tokyo 143-8555, Japan; 4Statistics and Decision Sciences, Global Development, Janssen R&D, A Division of Janssen Pharmaceutica NV, B-2340 Beerse, Belgium; 5Computational Biology & Toxicology, Preclinical Sciences and Translational Safety, A Division of Janssen Pharmaceutica NV, San Diego, CA 921921, USA

**Keywords:** hiPSC neurons, 2D, 3D, high throughput screening HTS, Ca^2+^ neuronal oscillations, neuronal active drugs, drug-induced seizure risk

## Abstract

Drug-induced seizure liability is a significant safety issue and the basis for attrition in drug development. Occurrence in late development results in increased costs, human risk, and delayed market availability of novel therapeutics. Therefore, there is an urgent need for biologically relevant, in vitro high-throughput screening assays (HTS) to predict potential risks for drug-induced seizure early in drug discovery. We investigated drug-induced changes in neural Ca^2+^ oscillations, using fluorescent dyes as a potential indicator of seizure risk, in hiPSC-derived neurons co-cultured with human primary astrocytes in both 2D and 3D forms. The dynamics of synchronized neuronal calcium oscillations were measured with an FDSS kinetics reader. Drug responses in synchronized Ca^2+^ oscillations were recorded in both 2D and 3D hiPSC-derived neuron/primary astrocyte co-cultures using positive controls (4-aminopyridine and kainic acid) and negative control (acetaminophen). Subsequently, blinded tests were carried out for 25 drugs with known clinical seizure incidence. Positive predictive value (accuracy) based on significant changes in the peak number of Ca^2+^ oscillations among 25 reference drugs was 91% in 2D vs. 45% in 3D hiPSC-neuron/primary astrocyte co-cultures. These data suggest that drugs that alter neuronal activity and may have potential risk for seizures can be identified with high accuracy using an HTS approach using the measurements of Ca^2+^ oscillations in hiPSC-derived neurons co-cultured with primary astrocytes in 2D.

## 1. Introduction

The incidence of drug-induced seizures is a significant concern in the development of new chemical entities (NCEs). Several cases of detection of such seizure liabilities, in late nonclinical and clinical development (leading to potential drug withdrawal from the market, e.g., Pentylenetetrazole), stress the need for better and more translational preclinical models that can be used early in the drug selection cascade [1,2]. Seizures, which can only be defined by recording an electroencephalogram (EEG) and may occur with or without external behavioral changes (such as convulsion), are sometimes difficult to detect in in vivo animal models. Irwin studies in mice [3] or rats [4] and dog models in early and late-stage nonclinical development are often referred to as reliable models to detect the potential of NCES for seizure/convulsive activity [5,6]. Moreover, a variety of in vitro assays using neuronal cells or brain tissue from animal origin have also been deemed applicable to detect drug-induced seizure liabilities [7,8,9]. However, it is well known that there may be species differences in sensitivity to drug-induced seizures among various preclinical animals, and there might also be some differences when compared to the outcome of humans [10,11,12]. In addition, in vitro assays using rat or mouse primary neurons and in vivo animal models are inefficient, with relatively low throughput, and expensive. Furthermore, given uncertainties about actual human translation, these approaches are not fully meeting the objectives of the 3Rs (Reduce, Replace, and Refine) concept of animal use. Recently, human induced pluripotent stem cell-derived neurons (hiPSC-neurons) have emerged as a promising human-derived neuronal in vitro platform that might be useful for the early de-risking phase of preclinical safety evaluations [13,14,15]. hiPSC-derived neurons co-cultured in 3D spheroids have been suggested to be amenable to phenotypic in vitro profiling use as well [16,17,18]. Applying such approach could increase throughput and reduce concerns about species differences between preclinical assays and outcomes in humans and, thus, be consistent with the 3Rs principles.

The multi-electrode array (MEA) technology has been applied to hiPSC neurons in in vitro assays but its relatively low throughput is a drawback of this approach in the early de-risking phase of drug development [13]. On the other hand, Ca^2+^-transient imaging using calcium-sensitive fluorescent dyes has been implemented for in vitro screening in a much higher throughput manner. In this approach, Ca^2+^ transients exhibit waveforms of synchronized neuronal Ca^2+^ oscillations and changes in the morphology and frequency of these phenomena may readily detect modulators of neuronal receptors and ion channels [19,20,21,22,23]. Intracellular Ca^2+^ levels that accompany neuronal depolarization in a single neuron had also been measured by imaging and evaluated but with severe limitations in the throughput similar to electrophysiological measurement approaches [24]. However, this limitation in throughput can be readily mitigated by establishing an HTS assay system based on a plate reader (in a 96- or 384-well format) using fluorescence calcium dyes [22,24].

Calcium fluorescence assays in a multi-well plate format, namely HTS screenings, allow the evaluation of hundreds to thousands of NCEs per year [23]. Although hiPSC-derived neurons exhibit changes in Ca^2+^ oscillations in responses to reference drugs in fluorescent-dye-based assays aimed to assess seizure hazard, the translational value of these assays to predict the potential seizure risk for NCEs has not been established. Therefore, in the present study, we used a straightforward approach to validate the translational value of the Ca^2+^ dye-based fluorescence assay using hiPSC-derived neurons cultured two-dimensionally (2D) with blind tests of 25 reference drugs that are known to be positive or negative for seizure risk in humans [25]. In this manner, we successfully managed to group reference drugs according to their altered neuronal activity (Figure 1B) or those that did not alter neuronal activity (Figure 1A). As expected, among the neuronally active drugs, we attempted to distinguish neuronal activity changes associated with drug-induced seizures from among the neuronally active drugs (Figure 1B).

Human 3D cerebral organoids could accurately recapture and mimic in vivo developmental stages of the brain tissues and might be more advantageous to study for future personalized medicine. However, the advantages of using 3D organoids over monolayer 2D culture for drug-induced toxic effects are still unclear. Therefore, we also investigated the effects of these 25 reference drugs on Ca^2+^ oscillations in spheroids made by three-dimensional (3D) cultures of hiPSC-derived neurons.

## 2. Materials and Methods

### 2.1. Human iPSC-Derived Neurons and Human Primary Astrocytes

In this study, we used cryopreserved excitatory neurons that were differentiated from human induced pluripotent stem cells (hiPSCs; Cat #: EX-SeV-CW50065) (Elixirgen Scientific, Inc., Baltimore, MD, USA). The hiPSCs used, namely, CW50065 were derived from a healthy 74-year-old female and distributed by the California Institute of Regenerative Medicine Human Induced Pluripotent Stem Cell Repository. Human iPSC-derived excitatory neurons were differentiated from CW50065 using a proprietary cocktail of differentiation-inducing factors called Quick-Neuron™ EX-SeV (Batch number: EX-SeV-CW50065-L; Elixirgen Scientific, Inc.), cryopreserved 3 days after differentiation induction and distributed. Human iPSC-derived excitatory neurons were recovered from cryopreservation and maintained essentially as instructed by the manufacturer.

Cryopreserved human primary astrocytes (#N7805100, Thermo Fisher Scientific, Waltham, MA, USA) were recovered as instructed by the manufacturer and seeded in a 10 cm dish at the density of 2,000,000 cells. The culture was maintained using the Human Astrocytes Kit (#N7805200, Thermo Fisher Scientific) for 12 days. Half of the culture medium was replaced with fresh medium every 2–3 days during the culture period. These astrocytes were harvested using StemPro™ Accutase™ Cell Dissociation Reagent (#A1110501, Thermo Fisher Scientific) when they reached confluency. These cells were cryopreserved in STEM-CELLBANKER^®^ GMP grade (ZENOGEN PHARMA, Kohriyama, Fukushima Japan) at the density of 2,000,000 cells per vial as working stocks for co-culture with neurons in 2D and 3D conditions described below. After a working stock was recovered, astrocytes were directly mixed with hiPSC-derived neurons for plating without further expansion.

The HiPSC line was obtained from the CIRM hPSC Repository funded by the California Institute of Regenerative Medicine (CIRM) with UC San Diego Approval- CW50065) [26]. The human Astrocytes Kit (commercial cells from Thermo Fisher Scientific) was used based on the applicable legal and ethical practices and guidance of uses of human subjects in Research in the United States (including 45 CFR Part 46, subparts A, B, C, D; Health Insurance Portability and Accountability Act; National Organ transplant Act −42 CFR Part 482; The Uniform Anatomical Gift Act (UAGA) of 1968, revised 1984).

### 2.2. Two-Dimensional Cell Culture

Before seeding cells, 384-well plates (#353962, Corning, Corning, NY, USA) were coated with 10 μg/mL of Laminin (#23017015, Thermo Fisher Scientific) diluted in PBS, 30 μL/well, at 37 °C for 2 h. Live neurons (hiPSC neurons) and human primary astrocytes were recovered from cryopreservation as described above and seeded in each well of 384-well plates at the density of 10,000 cells (HiPSC-cortical neurons) and human primary astrocytes at 2500 cells (ratio 4:1). These co-cultures were maintained under the medium conditions recommended by the manufacturer, namely Medium N(G2P), to complete the differentiation of excitatory neurons for the first week. Then, these cultures were maintained in Neurobasal^TM^ Plus (#A3582901, Thermo Fisher Scientific) supplemented with 2% B-27^TM^ Plus Supplement (#A3582801, Thermo Fisher Scientific), 1% GlutaMAX Supplement (#35050061, Thermo Fisher Scientific), 200 µM Ascorbic acid (#A4544, Sigma-Aldrich Inc., St. Louis, MO, USA), 1% Penicillin-Streptomycin (10,000 U/mL; #15140122, Thermo Fisher Scientific), 10% Neuron culture medium (#148-09671, FUJIFILM Wako, Osaka, Osaka, Japan), and 0.05% Component P (a mitotic inhibitor, Elixirgen Scientific Inc., Baltimore, MD, USA) for 5 weeks. Half of the culture medium was replaced with fresh medium every 3–4 days during the culture period.

### 2.3. Three-Dimensional Cell Culture

We used U-shaped-bottom 96-well plates (#174925, Thermo Fisher Scientific) without any prior coating to make a spheroid in each well. Neurons and astrocytes were seeded and maintained as described for 2D cultures, except for the following modifications. First, neurons and astrocytes were seeded in each well of the 96-well plates at the density of 8000 hiPSC neurons with 8000 human primary astrocytes (8000)/well (ratio 1:1). Second, when the differentiation of excitatory neurons was completed one week after seeding, cells were fed with BrainPhys^TM^ Neuronal Medium (#05793, STEMCELL Technologies, Vancouver, BC, Canada) supplemented with 2% SM1 Neuronal Supplement, 1% N2 Supplement-A, and 10% Neuron culture medium for 5 weeks.

### 2.4. Calcium Transient Spontaneous Oscillation Waveform Recording in 2D and 3D

Cells maintained under 2D conditions were incubated with 2 μM calcium-indicating dye, Cal-520 (#21131, AAT Bioquest, Sunnyvale, CA, USA), at 37 °C for 2 h as instructed by the manufacturer. After the incubation, the culture medium containing Cal-520 was replaced with 50 μL of HBSS supplemented with 0.1 mM MgCl_2_, 2 mM CaCl_2_, 10 mM HEPES, and 10 mM glucose and incubated at 37 ℃ for 15 min.

Cells maintained as spheroids (3D) in the U-shaped-bottom 96-well plates were transferred into 384-well spheroid plates (#4516, Merck, Corning, NY, USA) with 25 µL of the culture medium. A Ca^2+^-indicating dye solution including Fura-2 was prepared using the EarlyTox™ Cardiotoxicity Kit (#R8211, Molecular Devices, LLC, San Jose, CA, USA) according to the manufacturer. Then, the same volume of the dye solution was dispensed into each well. The pilot study did not support having good Ca^2+^ measurement to use Cal-520 dye.

Prior to the fluorescence measurements, all wells to be measured were treated with 0.04% dimethyl sulfoxide (DMSO) to stabilize the oscillation signal. The first recording started 10 min after the addition of DMSO for each plate for 20 min. The effects of the test compound were recorded again starting 60 min after the administration of compounds, for 20 min. In the 384-well plates, the wells at the edge of each plate (144 wells) were not used for adding test compounds or DMSO since the Ca^2+^ oscillations were not stable in the pilot experiments (the edge effects).

### 2.5. Compound Selection, Drug Dilution, and Addition

In the present study, 4-aminopyridine (4-AP at 30 µM) and kainic acid (KA at 10 µM) were used as positive controls, and aspirin (100 µM) and acetaminophen (100 µM) as negative controls. 4-AP is a K^+^ channel blocker that stimulates the release of both excitatory and inhibitory neurotransmitters in neurons both in vitro and in vivo and is known to increase glutamatergic transmission and cause seizures or convulsions [27]. KA is an excitatory toxic substance that stimulates the glutamate receptor and causes neuronal cytotoxicity and is often used in seizure models in vitro and in vivo [28,29,30]. Acetaminophen is a medication used to treat fever and pain with little incidence of seizure in humans with therapeutic doses.

Blinded samples of neat test drugs were sent to the testing site (RICOH Company Ltd., Tokyo, Japan) by Janssen Pharmaceutical NV (Beerse, Belgium) and stored at −20 °C until the day of testing. Four concentrations of each drug were studied (Table 1). The test concentrations were selected based on their respective free therapeutic C_max_ listed on PharmaPendium^®^ or based on the concentrations tested in our other in vitro assay [7].

Twenty-five compounds (Table 1) were first dissolved in either DMSO (most compounds) or distilled water depending on the compound, and further diluted with PBS to produce stock solutions concentrated 10-fold. Then, 5 μL of each compound solution was injected into each well using an automatic pipetting robot (#4505, ASSIST PLUS pipetting robot, Integra Biosciences Corp., Hudson, NH, USA), so that the final concentration of each compound was 1-fold nominal in the well. The final concentration of DMSO was adjusted to 0.1%. Changes in the fluorescence intensity (FI) produced by Ca^2+^ flux were measured at 37 °C for all wells of a 384-well plate simultaneously using the FDSS/µCell system (#C13299, Hamamatsu Photonics, Hamamatsu, Shizuoka, Japan). The excitation and the emission wavelengths were 470 nm and 540 nm, respectively. The recording duration was 20 min with a sampling frequency of 2 Hz for cells cultured under 2D conditions and 10 Hz for cells cultured under 3D conditions.

### 2.6. Data Analysis of Ca^2+^ Oscillation

Data obtained with FDSS/µCell were analyzed with IGOR Pro software (Wave Metrics, Portland, OR, USA) to derive five parameters including the peak number (pulses/10 min), peak width in ms (at 90% return to baseline), peak amplitude (AMP: RLU), the area under the curves, and the peak-to-peak time in ms (Figure 2A). For the detection of Ca^2+^ ion oscillation pulses, an original wave was differentiated after smoothing and pairs of bottom peaks and top peaks were detected from the differentiated waves by setting thresholds, which were counted as the number of pulses, and referred to as “peak number” further in the manuscript. The calcium oscillations count (=Peak number) was only taken from the large oscillation peak not from the small peak since the small peaks are too variable even in the negative control groups and interfere with the background noises.

The pulse width was calculated as the time duration between the bottom peak and the top peak. The background fluorescence intensity was estimated from the original wave amplitude by excluding the detected pulse regions and subtracting them from the original waves. The peak amplitude was calculated as the peak fluorescence intensity of the background-subtracted data at the pulse duration. The area under the curves was calculated by integrating the amplitude during the pulse duration. The peak-to-peak time was calculated as the time duration between successive peaks (Figure 2A). Data from baseline and 60 min after the test drug were used for statistical analysis. These parameters were also used by others [18].

The percent change at 60 min from baseline was analyzed separately for 2D and 3D assays. Nonparametric tolerance intervals of population coverage (P) = 80% and a confidence level (1−α) = 95% were constructed and reported for a vehicle (0.1% DMSO), a negative control (aspirin), and positive controls (kainic acid and 4-AP). For all test compounds, delta percent values were summarized by drug concentration on each plate. A nonparametric Wilcoxon test was used to compare the delta percent at a given concentration of a compound to DMSO data on the same plate. If the *p*-value from Wilcoxon test was below 0.05, a compound at a given concentration was considered statistically significantly different from DMSO. 

### 2.7. Sensitivity, Specificity, and Predictive Value of Potential Cardiac Risks in the Assay

Significant increase in the peak number observed in the Ca^2+^ oscillation was considered as a surrogate for drug-induced neuronal activity and, hence, potential seizure risks. Since some drugs cause seizure in humans at overdoses and/or in combination with other drugs, the predictivity values were calculated at free maximal plasma concentration in humans (fC_max_) and also at a threshold of 10-fold fC_max_ or 30-fold fC_max_. The list of reference drugs that have seizure events in humans was taken from the PharmaPendium^®^ (www.pharmapendium.com). A true positive (TP) was defined as neuronally active (or seizure potential risk) drugs that significantly increased the peak number, while a true negative (TN) was defined as non-neuronally active drugs in humans that did not significantly change the peak number at a tested concentration. A false negative (FN) was defined as a known drug that did not cause seizure in humans (<1% background incidence of seizure in men), while a false positive (FP) was defined as a non-neuronally active drug that significantly increased the peak number at >30-fold fC_max_. Sensitivity was calculated as the percent (%) of neuronal active drugs correctly predicted based on the significant increase in the peak number in this assay {TP/(TP+FN)}. Specificity was calculated as the percent of non-neuronally active drugs correctly predicted as non-seizure drugs {TN/(TN + FP)}. Positive predictive value (PPV) was calculated as PPV = {TP/(TP + FP)}, while negative predictive value (NPV) was NPV = {TN/(TN + FN)}.

### 2.8. Immunofluorescence

Cells maintained under 2D conditions were washed with PBS once and fixed with 4% paraformaldehyde solution in PBS (#161-20141, Fujifilm WAKO, Richmond, VA, USA) at room temperature for 15 min. After the fixation, cells were permeabilized with 0.1% Triton-X 100 (#X100, Sigma) in PBS at room temperature for 5 min. After three PBS washes, fixed cells were blocked with 3% BSA (#A9647, Sigma) in PBS at room temperature for 1 h. Blocked cells were incubated with a rabbit anti-GFAP polyclonal antibody (#Z0334, Dako, Agilent Technologies, Inc., Santa Clara, CA, USA) at 4 °C overnight. After three PBS washes, cells were further incubated with a mouse anti-MAP2B monoclonal antibody (#610460, BD Transduction Laboratories™, San Jose, CA, USA) as an additional primary antibody diluted 1:500 in 2% BSA in PBS at 4 °C overnight. After three PBS washes, cells were incubated with secondary antibodies, Alexa Fluor 488-conjugated goat anti-mouse IgG (#A11001, Thermo Fisher Scientific) diluted 1:2000 and Alexa Fluor 594-conjugated goat anti-rabbit IgG (#A11037, Thermo Fisher Scientific) diluted 1:500 in 2% BSA in PBS, protected from light, at room temperature for 1 h. After three PBS washes, cellular nuclei were stained with Hochest33342 (#H3570, Thermo Fisher Scientific) diluted 1:1000 in PBS at room temperature for 15 min. After three PBS washes, cells were imaged in PBS using a 20x objective lens of AxioObserver 7 (ZEISS; Munich, Germany).

Cells maintained as spheroids were transferred into a 1.5 mL tube, washed with PBS once, and fixed with 4% paraformaldehyde solution in PBS (#161-20141, Fujifilm WAKO) at 4 °C overnight with constant rotation (MTR-103, AS ONE Corporation, Osaka 550-0002, Japan). After two PBS washes, each spheroid (3D) was transferred into a Tissue-Tek Cryomold (#4565, Sakura Finetek Japan Co., Ltd., Tokyo, Japan) filled with Tissue-Tek O.C.T. compound (#4583, Sakura Finetek Japan Co., Ltd.). Cryomolds with single spheroids (3D) as such were sunk in liquid nitrogen. Cryosections of single spheroids of 20 μM thickness were obtained using a microtome-cryostat (CM1950, Leica, Wetzlar, Germany) at −20 °C and mounted onto glass slides (#MAS-01, MATSUNAMI GLASS, Osaka, Japan). The slides were dried at 4 °C overnight. The next day, the slides were warmed at room temperature for 30 min and soaked in PBS to remove O.C.T. compound. Sections of single spheroids were permeabilized with 0.1% Triton-X 100 (#X100, Sigma) in PBS at room temperature for 1 h and then blocked with 3% BSA (#A9647, Sigma) in PBS at 4 °C for 1 h after three PBS washes. Incubation of sectioned single spheroids with antibodies was essentially performed as previously described, except that the nuclei of sectioned spheroids were stained using Hochest33342 diluted 1:1000 in PBS at room temperature for 30 min. Sectioned spheroids were mounted with ProLong Glass (#P36982, Thermo Fisher Scientific) and cover glasses (#C024361, MATSUNAMI GLASS), dried at 4 °C overnight and imaged using a 20 “×” objective lens of AxioObserver 7 (Zeiss, Germany).

### 2.9. Characterization of Gene Expression (RNA-seq)

RNA was extracted from cells using the RNA 6000 Nano Kit (Agilent, Santa Clara, CA, USA) as per the manufacturer’s instructions. RNA quantity and quality were assessed using a NanoDrop spectrophotometer and Agilent 1000 bioanalyzer. cDNA libraries were prepared using a TruSeq Stranded Total RNA Library Prep kit and RNA-Seq was performed on the Illumina next-generation sequencing platform. Quantification of RNA-seq data: Transcript expression was quantified using the RNA-seq pipeline in Qiagen OmicSoft Studio V11.7. Target transcripts were derived from genome assembly Human.B38 from RefGene20210812. Transcript-level quantifications were transformed into gene-level count estimates. The counts were then normalized to a gene-level transcript per million (TPM) value which was then used as the gene-level abundance estimates used in the analysis.

## 3. Results

### 3.1. Time Course of Spontaneous Ca^2+^ Oscillations in hiPSC-Derived Neurons in 2D and 3D Cultures

In the pilot study, we observed that spontaneous Ca^2+^ oscillations were formed in hiPSC-derived neurons co-cultured with astrocytes in 2D (384-wells) and 3D (96-wells) after 4 weeks (DIV 32: days in vitro). The development of spontaneous intracellular calcium oscillations reached a plateau and stabilized at around 6 weeks in culture. The forming pattern of the oscillations among wells in both 2D and 3D cultures was less variable by 6–7 weeks, with most wells showing limited numbers of oscillations during 10 min recordings (Figure 1, baseline recordings). Although the number of oscillations per well was variable (<25 pulses/10 min), the rate was relatively constant during measurements within each well, which allowed the investigation of pharmacological interventions to modulate the frequency of oscillations during the period 42–49 IVD.

### 3.2. Measurement of Pharmacological Effects in hiPSC-Derived Neurons in 2D and 3D Cultures Using the Ca^2+^ Transient Assay

The significant responses to the positive controls 4-AP (30 µM) and kainic acid (100 µM) on Ca^2+^ transient oscillations were very clear and consistent in both 2D- and 3D-cultured hiPSC-derived neurons, while the negative control DMSO (0.1%) had little effect on Ca^2+^ oscillations (see examples in Figure 2B,C).

In Figure 1, we presented a simplified hypothesis of the relationship between excitatory and inhibitory responses to drugs on neuronal activity resulting in a “balanced” or “unbalanced” overall response. Drug-induced changes in the Ca^2+^ neuronal oscillations could thus be measured by significant decreases or increases, which were then used to identify neuronal active drugs and a subset of drugs with potential risks for seizure. Analyzing the data of drug-induced changes in Ca^2+^ neuronal oscillations with positive and negative controls as well as 25 reference drugs indicated that the most relevant parameter was the Ca^2+^ peak number. The peak number alone can be used to distinguish non-neuronal active drugs (no significant changes on calcium oscillations), neuronal active drugs, and neuronal active drugs with potential seizure risk. In the current study, the other four parameters (peak amplitude, area under the curves, peak-to-peak interval, and the peak width at 90% repolarization) did not add any additional value to the categorization of drug responses based on Ca^2+^ neuronal oscillations in 2D and 3D (Appendix A Appendix A). Therefore, we used the Ca^2+^ peak number in the present study as the basis for assessing the effects of drugs on Ca^2+^ neuronal oscillations and the performance of the assay in both 2D and 3D cultures.

An overview of the effects of the 25 reference drugs on Ca^2+^ neuronal oscillations based on Ca^2+^ peak number in 2D- and 3D-hiPSC-derived neurons co-cultured with human primary astrocytes is presented in Table 2 and Table 3. The predictive value for the detection of significant drug-induced significant changes or no significant changes in Ca^2+^ neuronal oscillations was 91% in 2D cultures and 45% in 3D cultures. The reasons for the differences in response between 2D- and 3D-hiPSC-derived neuron cultures for some drugs are unknown and will require further studies. However, in the present study, we have focused on the response of the 25 reference drugs in 2D-hiPSC-derived neurons co-cultured with human primary astrocytes.

The typical effects of 4 drugs on the changes of the peak number and some examples of Ca ^2+^ transient oscillation recordings are represented in Figure 3.

In hiPSC-derived neurons co-cultured with human primary astrocytes (2D format), the negative controls aspirin and amoxicillin did not significantly change Ca^2+^ neuronal oscillations. Acetaminophen, also a negative control, slightly but significantly increased Ca^2+^ neuronal oscillations only at 30 µM, which represents 65-fold its therapeutic fC_max_.

### 3.3. Analysis of Sensitivity, Specificity, and Balanced Accuracy

We analyzed the potential acute seizure risks of 25 reference drugs with known degrees of risk in humans based on significant changes in the peak number. Based on the numbers of true positives (TP), true negatives (TN), false positives (FP), and false negatives (FN), we calculated sensitivity (TP/(TP + FN)), specificity (TN/(TN + FP)), and balanced accuracy (TP + TN). The outcome is shown in Figure 4.

In summary, the study consisted of testing 22 neuronal-active and 3 non-neuronal-active drugs (25 reference drugs in total), covering both positive and negative controls. For potential seizure risk prediction in this assay, balanced predictivity was 91% with sensitivity and specificity of 85% and 100%, respectively (Figure 4). There were 0 out of 22 FP and 5 out of 22 FN compounds (Enoxacin, Bupropion, Clozapine, Pilocarpine, and pazopanib) in 2D hiPSC-derived neurons cultured with primary human astrocytes. On the other hand, the predictivity was much lower in 3D co-cultures (<40%).

### 3.4. Immunofluorescence

Immunofluorescence microscopy was carried out to confirm the presence of neurons and astrocytes in 2D co-cultures and 3D spheroids. We selected microtubule-associated protein 2 (MAP2) and glial fibrillary acidic protein (GFAP) as key markers for neurons and astrocytes, respectively, since MAP2 is highly and specifically expressed in mature neurons [31] and GFAP is expressed in mature astrocytes [32]. As is apparent in Figure 5, both 2D and 3D cultures showed intense fluorescence signals associated with the localization of MAP2 and GFAP with minimal to no overlap. Thus, we concluded that the 2D and 3D co-cultures contained both neurons and astrocytes after 42 days of in vitro culture (Figure 5).

### 3.5. Transcriptome Analysis of hiPSC-Derived Neurons Co-Cultured with Human Primary Astrocytes

To understand the characteristics of the hiPSC-derived neurons, co-cultured with human primary astrocytes, we identified the differences in gene expression between 2D cultures and 3D spheroids. Samples were taken from wells treated with 0.1% DMSO at the end of experiments. We assessed baseline gene expression levels of 48 key neuronal genes. The key neuronal neurotransmitter receptors and ion channels were highly expressed in both culture conditions (Figure 6). However, 92% of the genes showed greater baseline expression levels in the 3D spheroid compared to the 2D cultures. In particular, several genes (DLG4, GABRA1, GABAB3, GRA1,2,4, SLC17A6, and SYN1) showed much greater expression levels in the 3D spheroids compared to those in the 2D cultures (Figure 6).

## 4. Discussion

The present study shows that Ca^2+^-transient-analysis-based neuronal in vitro assays can be applied in early drug selection and optimization. Surprisingly, the data show that the 2D model we employed could help to significantly detect neuronal active drugs with seizurogenic potentials than the 3D model we used.

Development of human-based cellular HTS assays for use as a functional readout for drug-induced changes in neuronal network activity is still in the early stages and remains a challenge. However, a few HTS formats/technologies have been described. Measurement of spontaneous synchronized Ca^2+^ oscillations has been applied using rat cortical neuronal cultures [33] and hiPSC-derived neuronal cultures [16,19,34]. Calcium oscillations play a key physiological role in the nervous system. Spontaneous Ca^2+^ transients were reported to play an important role in the formation of functional synapses and rhythmic neurotransmitter release in neuronal networks in cultured neurons [33]. The hiPSC neurons co-cultured with human primary astrocytes used in the present study show the expression of genes linked to neuronal biological targets (receptors, ion channels, etc.) that are also expressed in the human adult brain. Modulation of gamma-aminobutyric acid (GABA) receptors, metabotropic glutamate receptors (mGluR) or the ionotropic glutamate (iGlu) receptor, N-methyl-D-aspartate (NMDA) receptors, permeable α-amino-3-hydroxy-5-methyl-4-isoxazole propionic acid receptor (AMPA receptor), kainate receptors, and delta receptor family, all contribute to the generation of Ca^2+^ oscillations in neurons [18,33,35]. Additionally, modulation of voltage-gated sodium channels, potassium channels, and Ca^2+^ channels, the ATP-gated P2X receptor cation channel family (P2X receptor), the superfamily of transient receptor potential (TRP) cation channels, and acid-sensing ion channels is also known to be associated with neuronal processes. Modulation of these channels and receptors by drugs can produce beneficial neuronal effects for the treatment of neuropsychological diseases and NCS diseases (so-called beneficial neuronal effects) but may also result in a risk for seizures at therapeutic concentrations or overdose [12,36]. It is critical to be able to distinguish between these beneficial therapeutic effects and neuronal toxicities that may result in seizures.

Following the assessment of the stability and the effects of drugs on multiple measurements of the Ca^2+^-transient oscillations, our data support the application of a single parameter peak number as the most relevant parameter for assessing changes in neuronal oscillations to distinguish neuronal-active from non-neuronally active drugs, and potential seizure risks among the neuronal-active drugs. The inclusion of other parameters characterizing Ca^2+^ transients to the analysis did not significantly improve the predictivity for the detection of drug-induced significant changes on Ca^2+^ oscillations based on the reference drugs tested (Appendix A Appendix A). This analysis approach provides a simplification of the assessment of the complex drug-induced effects on hiPSC-derived neurons. Other studies have used multiparametric approaches to analyze neuronal Ca^2+^ activities in neuronal cell cultures on a dish [12,15]. These parameters include peak number (count), peak amplitude, peak width, area under the curve, and peak-to-peak time of the Ca^2+^ transients. However, with multi-parametric analysis, the outcome and conclusions can be complex and potentially confusing. For example, the significant effects of a positive control, 4-AP, were found to be within the data cluster of DMSO controls in 3D-hiPSC-derived neuron cultures in a study reported by others [37]. Further studies with additional reference drugs encompassing additional pharmacological profiles will need to be evaluated using this HTS Ca^2+^-transient assay within hiPSC-derived neurons to confirm if a single parameter (i.e., the peak number) is sufficient and optimal for the determination of drug-induced changes in neural oscillations and potential risk for seizure.

In the present study, in 2D-cultured hiPSC-derived neurons, all neurotransmitters (acetylcholine, serotonin, glutamate, GABA, glycine, and norepinephrine) and respective reference drugs inhibiting these neurotransmitter receptors (e.g., bicuculline) or Sodium channel (phenytoin), Gap junction (mefloquine) [38], and selective norepinephrine reuptake inhibitors (maprotiline and amoxapine), or inhibitors of astrocyte activity (pazopanib—a potent and selective multi-targeted receptor tyrosine kinase inhibitor that inhibits astrocyte activity in vitro) [39]—all significantly reduced peak number. Pazopanib does not have a seizure risk in humans as it does not cross the blood–brain barrier. We used it as a reference drug to directly inhibit astrocyte activity in our in vitro assay. On the other hand, the following reference drugs with different pharmacological profiles significantly increased peak number: (1) cyclothiazide: a benzothiazide diuretic and antihypertensive that is a positive allosteric modulator of (AMPA)-type glutamate receptors [40]; (2) theophylline: a phosphodiesterase inhibitor used in therapy for respiratory diseases that has adenosine-receptor-blocking activity in the brain [41]; (3) 4-aminopyridine (4-AP): a neuronal voltage-dependent K^+^ channel inhibitor [42]; (4) clozapine: a psychiatric drug that modulates dopamine receptors with a high incidence of seizures in humans [43]; and (5) pilocarpine: a topical medication for glaucoma that modulates muscarinic receptors and is often used as a tool drug to induce seizure in vitro and in vivo [44].

Paracetamol, a drug not associated with seizure risk in humans, did not significantly reduce peak count within expected therapeutic exposure levels but did reduce this parameter at a very high concentration of 30 µM, representing ~65-fold its therapeutic-free C_max_. The clinical seizure rate incidence for paracetamol is <0.46%, which is within the background incidence in the human population [45]. Thus, we considered our findings with paracetamol at therapeutically irrelevant concentrations of 30 µM in 2D hiPSC neurons as not relevant.

Antibiotics, including fluoroquinolones (i.e., enoxacin, penicillin G, amoxicillin, and oxacillin), are reported to be associated with central nervous system toxicity and to trigger epileptic seizures via inhibition of gamma-aminobutyric acid (GABA-A) receptors as well activation of excitatory NMDA receptors [46,47]. However, the exact mechanisms for antibiotic-induced seizure in humans are not completely clear and may involve other more complex mechanisms, such as disturbance of neural protein synthesis, oxidative stress, and interactions in patients with pre-existing risk factors or co-medications such as antiseizure drugs [46]. Therefore, it may be more difficult to detect seizure risk with antibiotics using the current acute experiments in hiPSC-derived neurons in vitro. Bupropion is a dual norepinephrine and dopamine reuptake inhibitor that is used as an antidepressant medication and has been associated with seizure incidence (>2.8%) only in those taking overdoses of ≥600 to 900 mg/kg/day for days but has little incidence of seizure in those taking therapeutic doses of 10 or 50 mg/kg [48]. Moreover, bupropion was shown to attenuate kainic-acid-induced seizures in vitro [49]. Overdose levels of bupropion may lead to excessive tonic activation of a_1_/a_2_ adrenergic and 5HT_1A_ receptors by endogenous norepinephrine (NE) and 5-HT in DA neurons, which could be a mechanism for “delayed” seizure with a minimum onset of 24 h after the start of treatment [49,50]. Thus, the current acute assay in hiPSC neurons would not be expected to be able to detect this proposed seizure risk mechanism.

Interestingly, based on the 25 reference drugs, our analysis has shown that the predictive value for drug-induced changes in neural oscillations and potential seizure risk in 2D-cultured hiPSC-derived neurons (>90% predictive value) is much higher than that in 3D-cultured hiPSC-derived neurons (<40% predictive value). In 3D-cultured hiPSC-derived neurons, 15 of the 25 blinded reference drugs (bicuculine, glutamate, maprotiline, amoxapine, etc.) did not significantly change the peak number of oscillations. The reasons for the lower predictive value in 3D cultures with hiPSC-derived neurons are unknown and further studies will be required to understand the basis for this difference. Although key neuronal genes are expressed in both 2D- and 3D-cultured hiPSC-derived neurons, there are differences in expression levels between 2D and 3D neuron cultures for some genes. Since the numbers of cells between the two cell models were different, it is difficult to establish the relationship between these gene expression differences and the functional readout in the present study. Three-dimensional culture conditions have drawn the attention of many researchers recently because they are considered to be more representative of human organs and tissues. On the other hand, as evident in Figure 5, 3D spheroids used in the current study lack organized layered structures found in human organs. Moreover, they do not have any vasculatures, either. Perhaps the lack of organized structures and vascularization could be the primary reasons why 3D cultures of hiPSC-derived neurons did not perform well as in 2D culture.

There are clear limitations of the current study: the initial set of 25 reference drugs is relatively small and, thus, future studies with a larger and more diverse group of reference drugs with associated clinical data on seizure risk will be required for further validation. The designation of cut-off values of Ca^2+^ transient assay parameters that separate non-neuronally active drugs from neuronally active (changes in neural oscillations) or drugs with potential seizure risk will require further refinement based on future studies with a larger panel of reference drugs. The concentrations of positive controls (4-AP and KA) were too high in the current study, and these should be tested at lower concentrations in order to better define cut-off values for drugs increasing or decreasing Ca^2+^ oscillations. In addition, the variabilities of each of the measured parameters are larger when compared to other in vitro assays. Reducing this variability may require increasing the number of test wells per drug concentration, which could increase predictivity but reduce throughput. Additionally, the translational value of this Ca^2+-^ transient assay compared to other in vitro assays (i.e., MEA) and in vivo models for clinical relevance will need to be defined with future validation studies. Electrophysiology experiments are lacking here to further compare the observation observed in FDSS calcium oscillations. Finally, the current assay format can only detect potential changes in neuronal activity and seizure risk resulting from acute pharmacological changes. Delayed toxicities may involve complex mechanisms that may not be readily detectable using current in vitro assays, as discussed for some of the examples in drug responses. As such, potential delayed neuronal pharmacological or toxicological actions of a drug candidate may only be detected in long exposure to the testing compound or in the later stage in vivo safety models following chronic dosing in animals.

## 5. Conclusions

The discovery and development of novel and safe medicines is a long and expensive process, which is often associated with a considerable rate of attrition resulting in part from a safety concern. Therefore, in vitro assays, especially those that are human-based (e.g., hiPSC-derived neurons), may be capable of detecting safety issues (particularly seizure risk) earlier in discovery and development and could be extremely useful for early optimization and candidate selection, resulting in the reduction in human risk in clinical trials and approved marketed drugs. Future studies with a large panel of reference drugs will be needed to confirm and validate the utility of this HTS Ca^2+^-transient assay in 2D hiPSC neurons co-cultured with human primary astrocytes in 2D, along with further exploration of the 3D hiPSC neuron spheroid cultures, in which 3D cultures have human-like organized layered structures and vasculatures.

## Figures and Tables

**Figure 1 cells-12-00958-f001:**
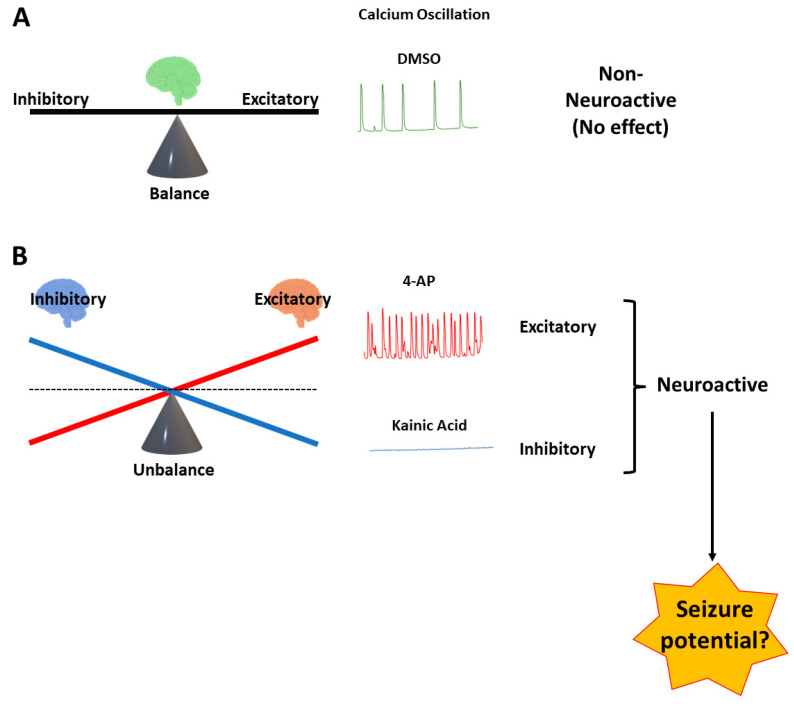
Schematic representation showing a simplified paradigm to understand the balance and the imbalance between excitatory and inhibitory neuronal activity based on changes in Ca^2+^ neuronal oscillations in hiPSC-derived neurons cultured with astrocytes. Upper panel (**A**) non-neuronally active response (e.g., 0.1% DMSO) shows no change in baseline oscillations; lower panel (**B**) neuronally active compounds can produce large increases in (e.g., 4-AP at 30 µM) or inhibition (e.g., kainic acid at 10 µM) of Ca^2+^ neuronal oscillations. Drug-induced large increases or decreases in Ca^2+^ neuronal oscillations, which tip the balance toward excitation or inhibition, are most likely involved in the potential risk for seizure activity.

**Figure 2 cells-12-00958-f002:**
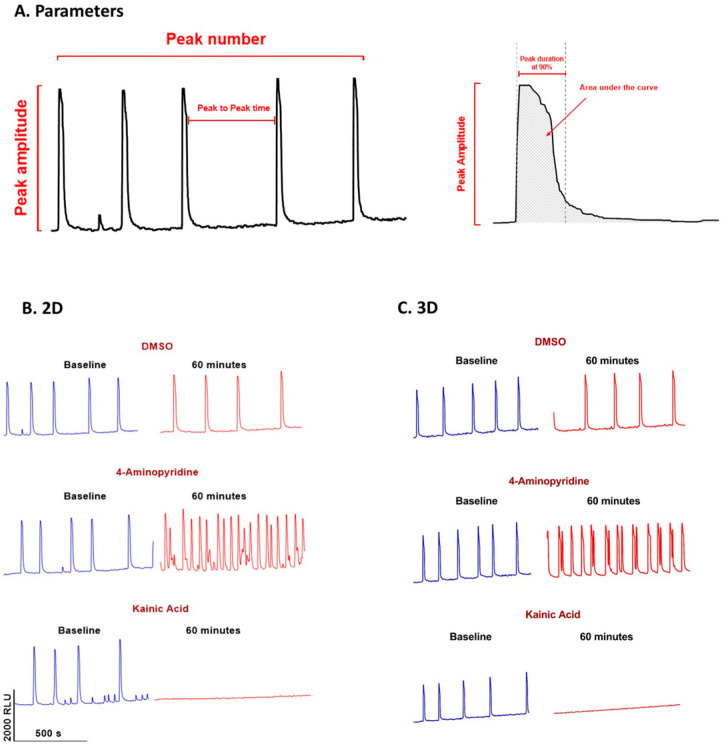
Examples of Ca^2+^-transient recordings. (**A**) Following parameters were measured as indicated: Ca^2+^ peak amplitude, peak count (peak number or account/10 min), pulse width, area under curves, and peak-to-peak interval. (**B**,**C**) The effects of 0.1% DMSO, 30 µM of 4-Aminopyridine and 100 µM of Kainic Acid on hiPSC-derived neurons cultured in 2D (**left**) and 3D (**right**) conditions with human primary astrocytes are shown. The traces are at around 60 min after the drug administration.

**Figure 3 cells-12-00958-f003:**
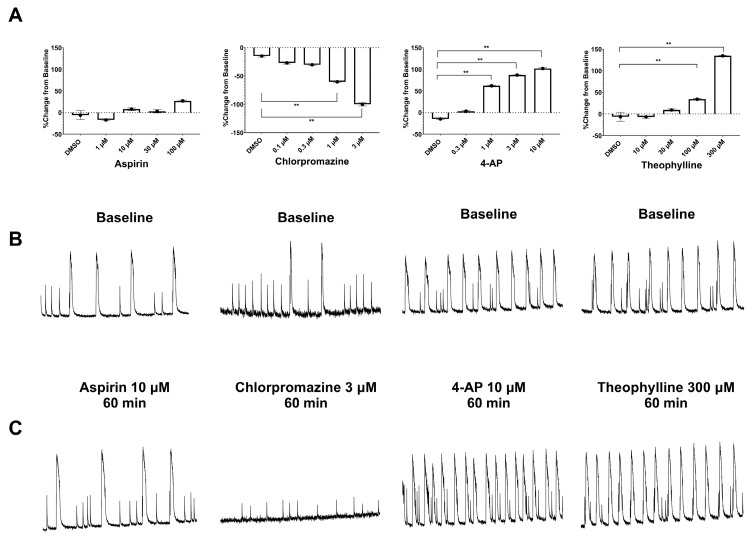
(**A**) Examples of concentration-dependent effects of chlorpromazine, 4-AP, and theophylline, compared to the negative control (aspirin), on the peak number of Ca^2+^ neuronal oscillations in hiPSC-derived neurons co-cultured with human primary astrocytes (2D format). **: *p* < 0.01 vs. 0.1% DMSO. (**B**,**C**) Examples of Ca^2+^ oscillations recorded at baseline (**B**) and 60 min after treatment (**C**) with aspirin, chlorpromazine, 4-AP, or theophylline in hiPSC-derived neurons co-cultured with human primary astrocytes (2D format).

**Figure 4 cells-12-00958-f004:**
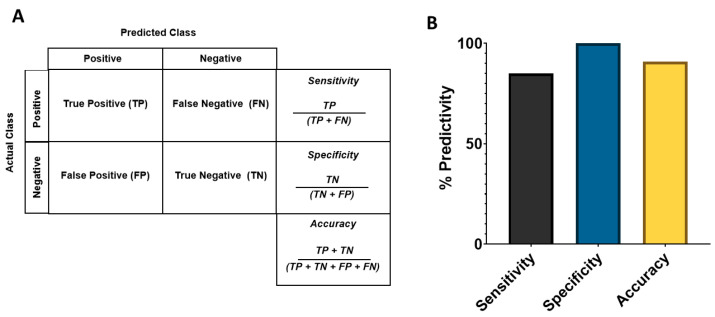
(**A**) Predicted class formulas. (**B**) Confusion matrix showing predictive values for hazard classification of drugs tested in the hiPSC-derived neurons co-cultured human primary astrocytes with FDSS assay. Sensitivity, specificity, and accuracy were derived from the true positive, false negative, false positive, and true negative classification of the 25 reference compounds in 2D hiPSC-derived neurons cultured with human primary astrocytes.

**Figure 5 cells-12-00958-f005:**
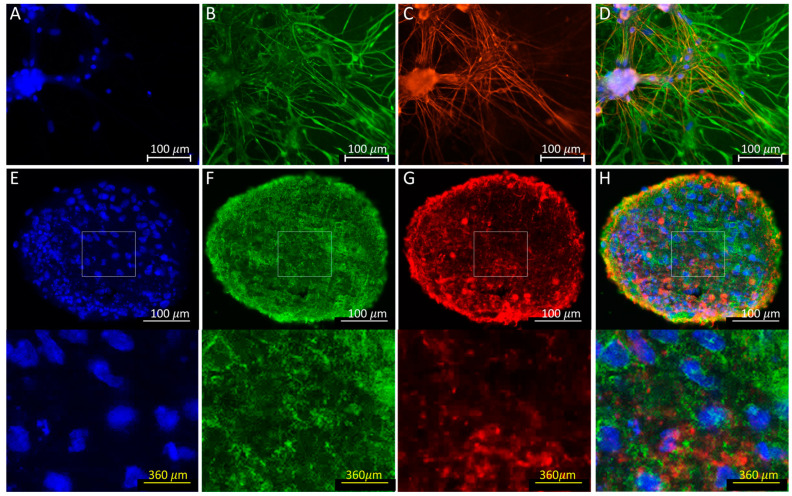
Immunofluorescence microscopy of hiPSC-derived neurons cultured with human primary astrocytes. Images show nuclei visualized using Hoechst 33342 (**A**,**E**) and localization of MAP2 (**B**,**F**) and GFAP (**C**,**G**) in the representative 2D (**A**–**D**) and spheroid (**E**–**H**) cultures. In D and H, the corresponding images (**A**–**C**) and (**E**–**G**) were merged, respectively. Rectangles shown in (**E**,**F**,**H**) are magnified at the bottom.

**Figure 6 cells-12-00958-f006:**
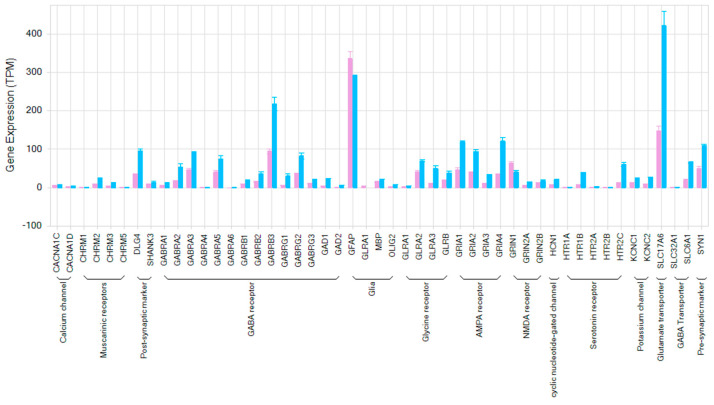
Bar plot showing in Transcript per Million (TPM) expression levels of 48 key selected neuronal genes in hiPSC-derived neurons with human primary astrocytes in 2D (pink) and 3D spheroids (blue).

**Table 1 cells-12-00958-t001:** The list of reference drugs tested in hiPSC-derived neurons co-cultured with astrocytes.

Compounds	Cas Number	Source	Free Cmax (µM)	Tested Concentrations (µM)
**DMSO**	-	Sigma	na	-
**Aspirin**	50-78-2	Sigma	0.04–0.16	3–100
**Amoxicillin**	26787-78-0	Sigma	0.003–32	0.1–3
**Acetaminophen**	103-90-2	Sigma	0.46	1–30
**Acetylcholine**	66-23-9	Sigma	1–10	10–300
**Serotonin**	153-98-0	Tocris Biosciences	1–2	3–100
**Bicuculline**	485-49-4	Sigma	na	0.3–10
**Glutamate**	56-86-0	Sigma	1–2	1–30
**Enoxacin**	74011-58-8	Sigma	1–4	0.3–10
**Cyclothiazide**	2259-96-3	Sigma	30	3–100
**Theophylline**	58-55-9	Sigma	8–15	10–300
**Maprotilene**	10262-69-8	Sigma	0.06	0.3–10
**Bupropion**	31677-93-7	Sigma	0.07	0.3–10
**Clozapine**	5786-21-0	Sequoia research products Ltd.	0.3	0.03–1
**4-Aminopyridine**	504-24-5	Sigma	0.025–0.075	0.3–10
**Amoxapine**	14028-44-5	Sigma	0.04–0.6	0.1–3
**Pilocarpine**	54-71-7	Sigma	0.03–0.55	0.3–10
**Chlorpromazine**	69-09-0	Sigma	0.05–0.15	0.1–3
**Pazopanib**	444731-52-6	Sigma	0.6–1.2	0.3–10
**Phenytoin**	57-41-0	Sigma	4–15	3–100
**Dizocilpine (MK-801)**	77086-21-6	Sequoia research products Ltd.	na	0.1–3
**Mefloquine**	51773-92-3	Sigma	0.095	1–30
**quinacrine**	69-05-6	Sigma	0.05	0.3–10
**Carbamazepine**	298-46-4	Sigma	10	3–100
**GABA**	56-12-2	Sigma	~1	0.1–3
**Kainic Acid**	487-79-6	Sigma	na	0.3–10

Cmax: Peak plasma concentration of drug administration in men; na: not available. Cas #: a unique identification number assigned by the Chemical Abstract Service. Sigma: Sigma-Aldrich, Belgium; Tocris Biosciences: Bristol, UK. Sequoia research products Ltd.: Berkshire, UK.

**Table 2 cells-12-00958-t002:** Overview of the summarized effects of 25 reference drugs on Ca^2+^ neuronal oscillations—Ca^2+^ peak number in hiPSC-derived neurons co-cultured with human primary astrocytes in 2D and 3D, relative to its free Cmax.

Compounds	Class of Drugs	Free Cmax (µM)	Ca²⁺ Transient Peak Number (2D)	Ca²⁺ Transient Peak Number (3D)
**Aspirin**	Negative control	0.04–0.16	± (100 µM)	± (100 µM)
**Amoxicillin**	Negative control	0.003	± (3 µM)	± (3 µM)
**Acetaminophen**	Negative control	0.46	± (10 µM)	± (10 µM)
**Bicuculline**	GABAA receptor antagonist	na	± (10 µM)	± (10 µM)
**Enoxacin**	Antibiotic	1	± (10 µM)	± (10 µM)
**GABA**	GABA receptor agonist	~1	↓ (1 µM)	± (3 µM)
**Cyclothiazide**	AMPA PAM	30	↑ (10 µM)	↑ (10 µM)
**Kainic Acid**	AMPA/NMDA/Glutamate receptor agonist	na	↓ (3 µM)	↑ (3 µM)
**Dizocilpine (MK-801)**	NMDA antagonist	na	↓ (0.1 µM)	↓ (0.1 µM)
**Glutamate**	Glutamate receptor agonist	1–2	↓ (1 µM)	± (30 µM)
**Theophylline**	Adenosine receptor antagonist	8–15	↑ (30 µM)	↑ (30 µM)
**Mefloquine**	Adenosine receptor A2a/Connexin43 antagonist	0.12	↓ (1 µM)	± (30 µM)
**Maprotiline**	Norepinephrine reuptake inhibitor	0.06	↓ (3 µM)	± (10 µM)
**Bupropion**	Norepinephrine–dopamine reuptake inhibitor	0.07	± (10 µM)	± (10 µM)
**Clozapine**	D2/5-HT2A antagonist	0.3	± (1 µM)	± (1 µM)
**Chlorpromazine**	Dopamine/5HTR receptor antagonist	0.05–0.15	↓ (0.3 µM)	± (3 µM)
**Amoxapine**	Reuptake of norepinephrine and serotonin inhibitor	0.04–0.6	↓ (3 µM)	± (3 µM)
**Serotonin**	5HTR agonist	1–2	↓ (3 µM)	↓ (3 µM)
**Acetylcholine**	Acetylcholine receptor agonist	1–10	↓ (30 µM)	↓ (30 µM)
**Pilocarpine**	Muscarinic acetylcholine receptor agonist	0.03–0.55	↑ (3 µM)	± (10 µM)
**4-Aminopyridine**	KCNA1 antagonist	0.025–0.075	↑ (1 µM)	↑ (1 µM)
**Phenytoin**	Sodium channel SCN1A antagonist	4–15	↓ (3 µM)	± (100 µM)
**Carbamazepine**	Sodium channel antagonist	10	↓ (30 µM)	± (100 µM)
**Quinacrine**	Phospholipase A2 inhibitor	0.05	↓ (10 µM)	± (10 µM)
**Pazopanib**	Tyrosine kinase inhibitor	0.6–1.2	± (10 µM)	± (10 µM)

±: no significant changes, Significant change (↓ or ↑): *p* < 0.05 vs. solvent in the same plate, is highlighted in red. Data were based on % changes of baseline value in the peak number. Cmax: highest plasma concentration in humans after the therapeutic dose. na: not available.

**Table 3 cells-12-00958-t003:** Effects of 25 reference drugs in increasing concentrations on average and standard deviation of the effect of Ca^2+^ peak number in hiPSC-derived neurons co-cultured with human primary astrocytes in 2D and 3D (expressed as % change from baseline).

		Peak Number—2D								Peak Number—3D							
Compound	Dose (µM)	Dose 1		Dose 2		Dose 3		Dose 4		Dose 1		Dose 2			Dose 3		Dose 4
		Mean	SD	Mean	SD	Mean	SD	Mean	SD	Mean	SD	Mean	SD	Mean	SD	Mean	SD
**4-AP**	0.37, 1.11, 3.33, 10	9.0	22.2	70.3 **	29.6	85.8 **	76.0	110.0 **	69.0	16.7	9.1	62.4 *	69.9	24.1 *	20.7	89.0 **	27.1
**Acetylcholine**	11.11, 33.33, 100, 300	2.4	26.9	−31.5	51.8	−47.2 *	38.9	−63.5 **	19.9	−33.8	23.9	−51.6 *	20.6	−34.9	38.3	−68.3 **	19.1
**Amoxapine**	0.11, 0.33, 1, 3	−5.2	8.5	25.3	51.8	12.5	27.9	−35.3 *	14.8	−10.1	16.0	−2.2	17.8	−10.8	18.4	−7.5	22.9
**Amoxicillin**	0.11, 0.33, 1, 3	−10.5	29.7	8.3	38.2	27.2 *	7.9	16.2	42.7	−8.0	8.8	1.4	17.2	9.3	35.0	18.3	28.6
**Aspirin**	3.7, 11.11, 33.33, 100	−10.2	14.5	17.6	21.1	10.1	32.9	27.9 *	35.6	−0.4	11.4	0.0	16.9	10.0	32.5	−8.2	17.6
**Bicuculine**	0.37, 1.11, 3.33, 10	−29.4	33.4	−3.5	30.0	9.9	60.8	5.8	53.6	−14.9	23.0	−13.0	23.0	5.8	35.2	−5.8	31.1
**Bupropion**	0.37, 1.11, 3.33, 10	12.7	16.6	6.6	34.9	−12.0	17.9	6.0	41.8	1.3	28.1	−8.4	9.1	−2.2	13.6	12.5	46.0
**Carbamazepine**	3.7, 11.11, 33.33, 100	−15.3	41.7	−17.0	21.0	−76.8 **	27.3	−100.0 **	0.0	11.4	12.9	3.9	24.7	−19.3	19.6	−21.5	40.8
**Chlorpromazine**	0.11, 0.33, 1, 3	−27.0	19.3	−29.5	6.7	−59.8 **	12.9	−100.0 **	0.0	6.5	32.0	−2.6	23.2	11.5	32.3	−4.8	15.4
**Clozapine**	0.04, 0.11, 0.33, 1	15.3	50.6	24.2	24.9 *	−15.0	32.3	−8.3	43.3	11.1	17.2	−10.0	14.9	−2.8	17.2	−13.9	16.4
**Cyclothiazide**	3.7, 11.11, 33.33, 100	2.7	30.9	178.3 **	116.3	−25.2	30.7	−82.0 *	33.4	0.0	27.6	21.5	33.0	63.6 *	50.0	102.9 **	50.5
**Enoxacin**	0.37, 1.11, 3.33, 10	1.3	26.3	2.3	32.6	5.5	22.6	2.0	17.3	−16.9	25.0	5.5	14.7	13.2	21.3	2.2	21.2
**GABA**	0.11, 0.33, 1, 3	−1.6	27.3	−4.4	34.6	−12.4	51.3	−36.5 *	34.7	5.0	23.4	−1.8	19.9	−0.6	28.3	2.2	24.8
**Glutamate**	1.11, 3.33, 10, 30	−9.3	60.3	−8.9 *	68.5	−60.3 *	49.9	−100.0 *	0.0	−7.0	20.1	12.3	33.6	20.0	34.4	8.4	17.3
**Kainic Acid**	0.37, 1.11, 3.33, 10	21.0	42.1	26.8 *	13.7	−53.3	43.8	−100.0 **	0.0	1.3	36.9	16.4	15.3	18.7 *	19.7	102.8 **	62.2
**Maprotiline**	0.37, 1.11, 3.33, 10	−0.7	18.1	0.5	39.7	−100.0 **	0.0	−100.0 **	0.0	−16.5	18.4	−8.2	15.9	−7.9	15.5	1.9	21.9
**Mefloquine**	1.11, 3.33, 10, 30	−30.8	21.7	16.2	60.2	−100.0 **	0.0	−100.0 **	0.0	2.8	21.5	−11.1	9.1	7.2	19.8	−3.7	26.5
**MK−801**	0.11, 0.33, 1, 3	−82.8 **	10.9	−90.4 **	9.3	−84.6 **	17.7	−96.3 **	5.7	−49.1 *	16.7	−67.3 **	24.4	−97.2 **	6.8	−84.8 **	9.4
**Paracetamol (acetaminophen)**	1.11, 3.33, 10, 30	−11.7	20.4	−25.0	32.0	−18.2	56.4	−94.5 **	13.5	2.2	37.2	−3.0	22.0	12.5	24.6	47.2 *	42.4
**Pazopanib**	0.37, 1.11, 3.33, 10	−49.0	31.8	4.4	27.5	11.3	60.3	10.0	54.1	6.4	38.1	−4.2	10.2	0.8	26.5	−2.0	36.4
**Phenytoin**	3.7, 11.11, 33.33, 100	−33.5	9.0	−15.2	14.9	−51.2 **	8.2	−27.5	14.4	1.4	18.4	−11.8	9.4	−14.6	11.9	−9.9	11.3
**Pilocarpine**	0.37, 1.11, 3.33, 10	11.5	18.4	8.3	36.2	34.3 *	41.0	−19.2	15.6	16.7	18.2	23.9	43.8	12.9	22.8	23.1	26.1
**Quinacrine**	0.37, 1.11, 3.33, 10	−2.7	27.8	−2.8	6.9	−11.0	55.8	−92.2 **	12.3	3.2	28.2	0.4	37.4	14.1	12.7	12.5	14.7
**Serotonin**	3.7, 11.11, 33.33, 100	−23.4	31.7	−1.7	74.7	−43.8	39.3	−45.8 **	22.5	−70.7 **	32.7	−69.1 **	29.8	−82.1 **	16.6	−80.3 **	29.4
**Theophylline**	11.11, 33.33, 100, 300	−5.7	22.7	24.7	43.3	39.3 *	23.5	118.4 **	105.4	2.0	14.7	53.5 *	60.4	78.1 *	80.0	114.2 *	98.1

*: *p* < 0.05 vs. 0.1% DMSO, **: *p* < 0.01 vs. 0.1% DMSO.

## Data Availability

The raw data supporting the conclusions of this manuscript will be available by the authors upon request without undue reservation.

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
