# Peer review of "High-Throughput Screening Assay for Detecting Drug-Induced Changes in Synchronized Neuronal Oscillations and Potential Seizure Risk Based on Ca2+ Fluorescence Measurements in Human Induced Pluripotent Stem Cell (hiPSC)-Derived Neuronal 2D and 3D Cultures"

_cells, 2023, doi:10.3390/cells12060958_

Round 1

Reviewer 1 Report

The manuscript by Lu et al provides a novel method for highthrouput testing of drugs for potential seizure risk in human induced pluripotent stem cells derived co-cultures of neurons with human astrocytes from a different source in 2D and 3D cultures using effects on synchronous calcium oscillations. The authors conclude that their 2D method is sensitive and accurate and can predict effects of drugs on potential seizure risk. In contrast, the 3D culture is far less accurate. The manuscript is interesting and potentially relevant but it has a number of shortcomings the limit its relevance. 

I dont understand what is the HiPSC line and what was it used for. 

First, the idea is not novel and has been previously developed in primary cultures of rat cortical neurons. The advantage here is the use of human induced pluripotent stem cells derived neurons co-cultured with astrocytes and the use of a plate reader. The authors should use calcium imaging to actually show that spontaneous and drug-induced calcium oscillations do occur synchronously and only in neurons as the mention. Once it is demostrated, the use of a plate reader is justified. 

The whole experiment should be amenable to be reproduced by others. Accordingly, a best description of the components of the Quick-Neuron solution and the exact procedure to differentiate stem cells into neurons is required. It is not clear to me what is the HiPSC line. 

Regarding calcium probes, a justification of the use of Cal520 as calcium probe should be included. If 2D and 3D systems are to be compared, then authors should use the same calcium probe or indicate they are not comparable. 

Presentation of the results with the parameter "peak number" is confusing.It is said that this value represents the number of peaks in 10 min, thus actually representing frequency of calcium oscillations. However, when peaks are much lower amplitude like in Fig. 2B (2D), are they counted as well? What means a negative value in the "peak number"? It could not be best expressed as % of the frequency of calcium oscillations?

2D and 3D cultures are kept for long periods of time from 4 to 10 weeks in vitro. It is expected that dring this time the fraction of astrocytes that proliferate in vitro are markedly different, potentially contributing to influence the results. Authors should test the proportion of astrocytes in their 2D and 3D cultures along culture time. 

Minors

Please define Cmax.

Spelling mistakes in lines 80 (aiumed), 129 (Reserach) and 563 (hiPSC-drevied)

Author Response

Authors’ responses: we thank for editor’s letter and reviewers’ comments concerning our manuscript. Those comments are all valuable and certainly improve the quality of manuscript. We looked into each comment carefully and made corrections. Our responses and corrections are as following. The manuscript are reviewed by native English senior scientists. 

Comments and Suggestions for Authors: Reviewer 1.

The manuscript by Lu et al provides a novel method for highthrouput testing of drugs for potential seizure risk in human induced pluripotent stem cells derived co-cultures of neurons with human astrocytes from a different source in 2D and 3D cultures using effects on synchronous calcium oscillations. The authors conclude that their 2D method is sensitive and accurate and can predict effects of drugs on potential seizure risk. In contrast, the 3D culture is far less accurate. The manuscript is interesting and potentially relevant but it has a number of shortcomings the limit its relevance. 

I dont understand what is the HiPSC line and what was it used for. 

Authors’ response: The aims and novelty of the uses of HiPSC-neurons are explained clearly in Introduction line 65-75.

We used a human induced pluripotent stem cell (hiPSC) line generated by California Institute for Regenerative Medicine (CIRM) and distributed by Fujifilm CDI. The line ID is CW50065 and it was used to differentiate excitatory neurons. This statement has been summarized and can be found on lines 111-115 of the main text.

First, the idea is not novel and has been previously developed in primary cultures of rat cortical neurons. The advantage here is the use of human induced pluripotent stem cells derived neurons co-cultured with astrocytes and the use of a plate reader. The authors should use calcium imaging to actually show that spontaneous and drug-induced calcium oscillations do occur synchronously and only in neurons as the mention. Once it is demonstrated, the use of a plate reader is justified. 

Authors’ response: The calcium oscillations are clearly demonstrated in Figure 1-3. Although the use of plate readers was referenced in the Introduction section of this manuscript, our calcium assay was carried out based on imaging using Functional Drug Screening System, namely FDSS, as stated on line 207 of the manuscript. In addition, based on our preliminary studies, we could not observe spontaneous calcium oscillation in astrocyte monocultures. Therefore, we concluded that the spontaneous calcium oscillation detected in neuron-astrocyte co-cultures occurred only in neurons. We will state this finding in the Materials and Methods section.

The whole experiment should be amenable to be reproduced by others. Accordingly, a best description of the components of the Quick-Neuron solution and the exact procedure to differentiate stem cells into neurons is required. It is not clear to me what is the HiPSC line. :

Authors’ response: Excitatory neurons were differentiated from hiPSCs using Sendai-virus mediated forced expression of neuron-inducing transcription factors. Reagents used and the differentiation protocol have been publicly available through the following link: https://www.elixirgensci.com/ipsc-derived-cells-and-differentiation-kits/excitatory-neurons/

The differentiation protocol itself has been applied to dozens of hiPSC lines or to CW50065 to produce almost 100 different lots of cryopreserved excitatory neurons, thereby being reproducible. This statement will be added in the Materials and Methods section. As for the explanation of “HiPSC line”, please see our response above.

Regarding calcium probes, a justification of the use of Cal520 as calcium probe should be included. If 2D and 3D systems are to be compared, then authors should use the same calcium probe or indicate they are not comparable. Rie please answer

Authors’ response: As stated on line 176, we have concluded based on our preliminary study that the use of Cal-520 did not produce reliable Ca2+ oscillation in spheroids

Presentation of the results with the parameter "peak number" is confusing. It is said that this value represents the number of peaks in 10 min, thus actually representing frequency of calcium oscillations. However, when peaks are much lower amplitude like in Fig. 2B (2D), are they counted as well? What means a negative value in the "peak number"? It could not be best expressed as % of the frequency of calcium oscillations?

Authors’ response: The peak numbers were counted only for strong peaks in 20 min by differentiating the signals and thresholding the differentiated waves, thus weak peaks were eliminated for counting. The numbers in Table 2 should be a change of peak number from baseline as pointed out by Reviewer 1. Revised Table 2 will be submitted.

We now clearly explained what is the peak number without small peak, indeed in line 253-255.There are no negative value on the peak number: was expressed as % changes of the baseline value: ±: no significant change, ↑ or ↓: significant increase or decrease. Negative control: means that the tested drug has no seizure or neuronal effects in human. Now added this and also Cmax into the legend of the Table 2.

2D and 3D cultures are kept for long periods of time from 4 to 10 weeks in vitro. It is expected that dring this time the fraction of astrocytes that proliferate in vitro are markedly different, potentially contributing to influence the results. Authors should test the proportion of astrocytes in their 2D and 3D cultures along culture time.

Author’s response: to neurons appears to be consistent with the seeding condition. This will be stated in the Materials and Authors’ response: We concluded that astrocytes do not proliferate under the conditions described in the manuscript because we did not observe increase in cell density in 2D culture and spheroid size in 3D culture over the duration of culture. ICC images shown in Figure 5 were derived from cultures 6 weeks after seeding cells and the proportion of astrocytes Method section

Minors

Please define Cmax.

Spelling mistakes in lines 80 (aiumed), 129 (Reserach) and 563 (hiPSC-drevied)

Author’s response: All corrected now.

Reviewer 2 Report

This study aims at establishing a high-throughput screening assay to evaluate the drug impact on intracellular calcium handling in hiPSC-derived neurons co-cultured with astrocytes. They compared the 2D and 3D cultures. The work is original and could contribute to the field. However, improvements are needed. Please see my major comments below:

Methods:

Why did you pick iPSC line from elderly women? More information about the healthy control should be specified.

How did you improve the maturation of the neurons in this study? What are the readouts?

The authors claim that the neurons are excitatory. Did they verify this claim? How? What kind of neurons do you obtain, i.e., central vs peripheral? Drug effects may change. Please justify.

The authors should justify the choice of the the medium for neurons and its compatibility with astrocytes.

Please explain why the medium for maturation differ between 2D and 3D cultures.

The experiments last for 60 min at least. Were they performed at RT? Temperature can modulate the intracellular calcium handling. Please clarify.

Results:

3.1

What does IVD mean?

Sometimes neurons are not spontaneously excitable. How the authors discriminate these neurons with others exhibiting spontaneous oscillations? What is the ratio in this work eventually?

3.2

There is no graph about aspirin in figure 2.

How do you discriminate the following parameters: neuronal active drugs vs neuronal active drugs with potential seizure risk?

Do the author take in account basal cytosolic oscillations corresponding to calcium leak from ER?

The table provided lacks explanation. It is unclear. Please clarify.

3.3

What does FDSS mean ?

3.4 

Considering that MAP2 can be also expressed in astrocytes, another marker to stain the neurons should be used.

Discussion :

How to explain the differences observed in 2D and 3D?

Electrophysiology experiments are lacking to further complete the observation observed with calcium oscillations. Somehow, seizure remains an electrical manifestation.

Author Response

Reviewer 2:

Comments and Suggestions for Authors

This study aims at establishing a high-throughput screening assay to evaluate the drug impact on intracellular calcium handling in hiPSC-derived neurons co-cultured with astrocytes. They compared the 2D and 3D cultures. The work is original and could contribute to the field. However, improvements are needed. Please see my major comments below:

Response: 

We thank Reviewer 2 for their positive feedback

Methods:

Why did you pick iPSC line from elderly women? More information about the healthy control should be specified.

How did you improve the maturation of the neurons in this study? What are the readouts?

The authors claim that the neurons are excitatory. Did they verify this claim? How? What kind of neurons do you obtain, i.e., central vs peripheral? Drug effects may change. Please justify.

The authors should justify the choice of the the medium for neurons and its compatibility with astrocytes.

Please explain why the medium for maturation differ between 2D and 3D cultures.

The experiments last for 60 min at least. Were they performed at RT? Temperature can modulate the intracellular calcium handling. Please clarify.

Authors’ response: this hiPSC line, CW50065, was derived from a 74-years old female. Since this individual remained healthy at the time of blood donation, the hiPSC line derived from this individual was listed as the hiPSC line that can be used as a control for research. Additionally, this hiPSC line can differentiate very well into a variety of cell types in all three germ layers. These are the reasons why we use this hiPSC line for the current study. This statement will be added to the Materials and Methods section.

How did you improve the maturation of the neurons in this study? What are the readouts?

Authors’ response: It is beyond the scope of this study to include all of our preliminary data. However, preliminarily we have optimized the culture of hiPSC-derived excitatory neurons mixed with human primary astrocytes in a 2D format using the multi-electrode array (MEA) system and in a 3D spheroid format using FDSS-Ca2+ system. Based on the periodical appearance of spontaneous network bursts, we concluded that the culture conditions and the duration of culture before measurement applied in the current study are the best.

The authors claim that the neurons are excitatory. Did they verify this claim? How? What kind of neurons do you obtain, i.e., central vs peripheral? Drug effects may change. Please justify.

Authors’ response: Based on the transcriptomic analysis by RNA-seq, these neurons are best described as glutamatergic neurons since they highly express relevant markers including GLS, GRIN2B and SLC17A6. Consistent with the transcriptomic analysis, their responses to known drugs indicate that they have excitatory synapses based on the MEA system. However, we were unable to determine whether these neurons represent the central nervous system or peripheral neurons.

The authors should justify the choice of the the medium for neurons and its compatibility with astrocytes.

Please explain why the medium for maturation differ between 2D and 3D cultures.

Authors’ response: As stated above, these media conditions were identified best among tested conditions based on the MEA system.

The experiments last for 60 min at least. Were they performed at RT? Temperature can modulate the intracellular calcium handling. Please clarify.

Authors’ response: During recording of calcium transient spontaneous oscillation waveform cultures were kept at 37℃ and after the drug administration they were also maintained at  37℃ until recording. This will be stated in the Materials and Methods section.

Results:

3.1

What does IVD mean?

Sometimes neurons are not spontaneously excitable. How the authors discriminate these neurons with others exhibiting spontaneous oscillations? What is the ratio in this work eventually?

Author’s’ response: we now changed IVD into DIV: days in vitro

First of all, with calcium dye-based imaging used in this study, one can observe neuronal network bursts only, unlike MEAs with which one can record spontaneous random firing of neurons, too. For this study, we examined all wells in 384-well plates before administration of any drug to detect basal activity of neurons. The fraction of wells that showed at least two Ca2+ oscillations was 89.5% in 2D cultures and 99.7% in 3D spheroid cultures. Wells showing one or no Ca2+ oscillation were excluded from the analysis and the fraction of wells showing no baseline activity was 2.1% in 2D cultures (19/912) and 0.33% (3/912) in 3D spheroid cultures. This will be stated in the Materials and Methods section. On the other hand, at the cellular level, we could not distinguish neurons that are excitable spontaneously from the ones that are not under current experimental setting and it is beyond the scope of this study to distinguish these two neuron populations at a cellular level.

3.2

There is no graph about aspirin in figure 2.

How do you discriminate the following parameters: neuronal active drugs vs neuronal active drugs with potential seizure risk?

Do the author take in account basal cytosolic oscillations corresponding to calcium leak from ER?

The table provided lacks explanation. It is unclear. Please clarify.

3.3

What does FDSS mean ?

 Authors’ response: it stands for Functional Drug Screening System.

3.4 

Considering that MAP2 can be also expressed in astrocytes, another marker to stain the neurons should be used.

Authors’ response: Given that the authors’ responses need to be submitted within 10 days and we do not currently have any cultures similar to ones described in this manuscript, we are unable to test another marker to stain neurons. On the other hand, the combination of anti-MAP2 and GFAP antibodies to distinguish neurons from astrocytes has been used in many other publications: e.g., Int J Mol Sci (2021) 22(23):12770; J Neuroinflammation (2018) 15(1):294; Cells (2021) 10(10): 2683; J Neurosci (1987) 7(10):3163-3170; Hippocampus (2016) 26(8):980-94; J Neurochem (2005) 95(5):1421-37; J Neurochem (1998) 71(4):1421-8. Furthermore, human primary astrocytes used in this study was stained with an anti-MAP2 antibody in our preliminary study and we have confirmed that they express MAP2 at a detection limit level:

Discussion :

How to explain the differences observed in 2D and 3D?

Electrophysiology experiments are lacking to further complete the observation observed with calcium oscillations. Somehow, seizure remains an electrical manifestation.

Authors’  response:

in line 590-608: we explained the possible difference why 3D culture had less detectable value.

We added “ Electrophysiology experiments are lacking here to further compared the observation observed with the data from the calcium oscillations” in the limitation of the study (line 623-624) according to the comment.

Reviewer 3 Report

This is an interesting and important study of developing a high throughput assay for seizure risk of a list of 25 drugs with known clinical seizure incidence.

Comments:

The comparison of 2D and 3D cultures was useful, but what was the rationale for this comparison? A comment regarding future use of 3D human cerebral organoids would be important. Also mention of the possibility of using this technology for personalized medicine, deriving iPSCs from affected patients would be worthwhile.

Fig. 1. A time calibration is needed.

Fig. 2A and B. Add an RLU/time base calibration.

Table 2. It is unclear as to what is meant by Ca transient peak number. The only number is a concentration, which is confusing. Please explain.

Overall a thorough discussion and well referenced.

Author Response

The comparison of 2D and 3D cultures was useful, but what was the rationale for this comparison? A comment regarding future use of 3D human cerebral organoids would be important. Also mention of the possibility of using this technology for personalized medicine, deriving iPSCs from affected patients would be worthwhile.

Fig. 1. A time calibration is needed.

Fig. 2A and B. Add an RLU/time base calibration.

Table 2. It is unclear as to what is meant by Ca transient peak number. The only number is a concentration, which is confusing. Please explain.

Overall a thorough discussion and well referenced.

Author’s Responses:

We thanks for the reviewer’s positive feedback.

according to the comment, we added a few lines for this 2D vs 3D rational comparison and mention of advantage of uses of 3D in the introduction.

Fig 1 and Fig 2 : time added now.

Calcium transient peak number is now explained in Figure 2 and in the text ( Line 245-247).

Table 2: now explained in the legend.

Round 2

Reviewer 1 Report

The authors have addressed satisfactorily the questions and concerns.

If authors have, in any way, received payment (salary or others) by the companies that have apparently supported finantially this work they should mention it as a conflict of interest.